# Simple and Highly Efficient Detection of PSD95 Using a Nanobody and Its Recombinant Heavy-Chain Antibody Derivatives

**DOI:** 10.3390/ijms24087294

**Published:** 2023-04-14

**Authors:** Markus Kilisch, Maja Gere-Becker, Liane Wüstefeld, Christel Bonnas, Alexander Crauel, Maja Mechmershausen, Henrik Martens, Hansjörg Götzke, Felipe Opazo, Steffen Frey

**Affiliations:** 1NanoTag Biotechnologies GmbH, Rudolf-Wissell-Straβe 28a, 37079 Göttingen, Germany; 2Synaptic Systems GmbH, Rudolf-Wissell-Straβe 28a, 37079 Göttingen, Germany; 3Institute of Neuro- and Sensory Physiology, University Medical Center Göttingen, 37073 Göttingen, Germany; 4Center for Biostructural Imaging of Neurodegeneration (BIN), University of Göttingen Medical Center, 37075 Göttingen, Germany

**Keywords:** PSD95, PSD-95, SAP-90, nanobody, single-domain antibody, sdAb

## Abstract

The post-synaptic density protein 95 (PSD95) is a crucial scaffolding protein participating in the organization and regulation of synapses. PSD95 interacts with numerous molecules, including neurotransmitter receptors and ion channels. The functional dysregulation of PSD95 as well as its abundance and localization has been implicated with several neurological disorders, making it an attractive target for developing strategies able to monitor PSD95 accurately for diagnostics and therapeutics. This study characterizes a novel camelid single-domain antibody (nanobody) that binds strongly and with high specificity to rat, mouse, and human PSD95. This nanobody allows for more precise detection and quantification of PSD95 in various biological samples. We expect that the flexibility and unique performance of this thoroughly characterized affinity tool will help to further understand the role of PSD95 in normal and diseased neuronal synapses.

## 1. Introduction

The post-synaptic density (PSD) is a characteristic, disc-shaped, and electron-dense region lining the postsynaptic membrane of excitatory synapses (Figure 1A). Functionally, it plays a crucial role as an organizational scaffold, orchestrating the recruitment of neurotransmitter receptors, ion channels, and corresponding signaling molecules [1,2]. The precise molecular composition and organization of the PSD are expected to have a large impact on the post-synaptic response upon neurotransmitter release from the pre-synapse. Vice versa, the composition and size of the PSD may differ depending on the localization within the brain, the activity and maturation state of individual synapses, and the response to neurodegenerative diseases [3,4].

PSD95 (post-synaptic density protein 95) is a prominent component of the PSD and the best-studied member of the Membrane-Associated Guanylate Kinase (MAGUK) family of proteins, sharing a characteristic organization of PDZ, an Src homology domain (SH3), and a Guanylate Kinase (GK) domain [5] (Figure 1B). PSD95 forms heterodimeric complexes with the related MAGUK family member PSD93. Several hundred copies of both proteins form a multimeric scaffold [2,6,7] that is a major constituent of the post-synaptic density in neurons. PSD95 plays an important role in synaptic strength plasticity and the stabilization of synaptic changes during long-term potentiation. Recently, there has been overwhelming evidence correlating PSD95 disruption with cognitive and learning deficits observed in schizophrenia and autism [8].

Due to its functional relevance in synaptic plasticity and its well-established localization, PSD95 is a common marker used for the specific immunolabeling of the postsynaptic side of excitatory synapses in the central nervous system [9,10]. However, reliable immunodetection of PSD95 using conventional antibodies tends to be notoriously difficult. The most likely cause is the compact and protein-dense structure of the PSD, which impairs the access of antibodies to their respective epitopes, especially when samples are chemically fixed. To allow immunodetection of PSD95, dedicated protocols have been developed that use non-fixed fresh-frozen material or rely on mild fixation and/or additional antigen retrieval steps, which may even involve proteolytic treatment [11]. While such protocols generally allow for decent immunodetection of PSD95, they may be suboptimal for other applications. This, e.g., often applies if a co-staining of other target proteins is intended. In such cases, the other target proteins or the antibodies used for their detection may not be compatible with the special conditions required for immunostaining of PSD95. Ultimately, these differences may have a major impact on the reproducibility of staining experiments and thereby greatly complicate data analysis.

We speculated that most of these problems might be caused by the densely crowded environment around PSD95, allowing only suboptimal access to large probes like conventional antibodies. If that holds true, smaller affinity probes may facilitate immunostaining of PSD95 and ideally also circumvent the need for any special specimen preparation. Camelids like alpacas are equipped with a special set of immunoglobulins without light chains, known as heavy-chain only antibodies (hcAbs). The antigen-binding variable heavy-chain domains of hcAbs (VHH) can be individually produced as single-domain antibodies (sdAbs, also known as nanobodies) in bacteria [12] as well as in mammalian cells [13]. With only ~15 kDa, they are significantly smaller than a conventional immunoglobulin (~150 kDa), but nevertheless possess the complete information and potency for binding its target strongly and specifically.

In this study, we characterize a high affinity nanobody binding PSD95. A directly labeled fluorescent version of this recombinant single-domain antibody efficiently enters the PSD and thereby enables a bright and specific one-step immunofluorescent detection of PSD95 on PFA-fixed samples without the need for any special treatment. Importantly, we also generated Fc fusions of this nanobody (recombinant heavy-chain antibody; r-hcAb), resulting in affinity probes that are fully compatible with commonly used heat-induced antigen retrieval protocols performed on tissue and specifically recognize PSD95 in Western blot applications.

## 2. Results

To generate anti-PSD95 nanobodies, two alpacas were immunized with a recombinant protein fragment comprising PDZ-domains 1 + 2 of mouse PSD95, showing only two conservative amino acid exchanges with respect to the corresponding human and rat sequences (Figure 1B,C). After immunization, total RNA extracted from peripheral blood lymphocytes was reversely transcribed to generate nanobody-specific cDNA that was subsequently cloned into a pHen2-derived phagemid. After three rounds of phage display, 96 individual nanobody clones were expressed in *E. coli* and initially analyzed for interaction with the immunized antigen by ELISA.

The anti-PSD95 nanobody clone 1B2 (“NbPSD95”) displayed the strongest colorimetric signal against the immunized antigen in the initial single-clone ELISA and was analyzed regarding its cross-reactivity towards rat PSD95 and related PDZ-domain-containing proteins in two experimental setups: First, the PDZ-domains 1 and 2 (PDZ1/2) from rat PSD95 (RtPSD95), PSD93 (RtPSD93), SAP102 (RtSAP102) or SAP97 (RtSAP97; see Figure 1C) were recombinantly expressed and purified as fusions to *E. coli* maltose-binding protein (MBP) and subsequently coated on ELISA plates. A dilution series of FLAG-tagged NbPSD95 was used to determine its binding preference to the coated PDZ domain-containing protein fragments. In this assay, a significant interaction of NbPSD95 was only detected for proteins harboring the PSD95-derived PDZ domains (RtPSD95^PDZ1/2^), showing that in vitro the nanobody specifically recognized PSD95 but none of the corresponding PDZ-domains from the other proteins analyzed (Figure 2A).

In a second assay (Figure 2B), COS-7 cells were transiently transfected with the different PDZ1/2 fragments fused to EGFP and a mitochondrial targeting sequence derived from TOM70. This resulted in the display of the various PDZ1/2 domain fragments on the surface of the mitochondria, exposed to the cytoplasm. Cells were fixed with 4% PFA and stained with NbPSD95 conjugated directly with a fluorophore (also referred to as “FluoTag-X2 anti-PSD95”). In this setup, NbPSD95 only stained mitochondria when the expressed GFP fusion protein comprised PDZ-domains derived from PSD95, thereby corroborating the specificity of NbPSD95 for its target. Importantly, the performed PFA fixation did not interfere with NbPSD95 binding, indicating that recognition of the target structure is not critically affected by modification of primary amines. In this context, it is interesting to note that, with some few exceptions, almost all lysine residues (K) are highly conserved between PSD95, PSD93, SAP97, and SAP102 (Figure 1C).

To provide more versatility to this NbPSD95 and allow its detection using common secondary reagents, the nanobody was fused in frame to Fc domains from either rabbit IgG or mouse IgG1 or IgG2, thereby creating various NbPSD95-Fc chimeras. The resulting recombinant heavy-chain antibodies (r-hcAbs) anti-PSD95 could be expressed in HEK-293 cells and successfully detected PSD95 in Western blot applications when tested on mouse brain lysates together with secondary reagents recognizing either rabbit or mouse IgGs (Figure 2C and Appendix A). Importantly, the specific signal was completely absent when using a lysate prepared from PSD95 knock-out mice, corroborating the specificity of the nanobody used as the antigen-binding moiety.

To see if NbPSD95 would also be able to detect PSD95 within bona fide postsynaptic structures, primary hippocampal neuronal cultures from rats were fixed with 4% PFA and stained with fluorescently labeled NbPSD95. As expected for PSD95, a bright punctate and neuron-specific staining was observed, indicating that NbPSD95 could indeed detect PFA-fixed PSD95 in the postsynaptic density of neurons (Figure 2D).

Encouraged by these results, we set out to directly compare our anti-PSD95 tools (NbPSD95 and an anti-PSD95 r-hcAb) to the most-cited anti-PSD95 antibody (Millipore MABN68; clone K28/43; mouse IgG2) (Figure 3 and Appendix A). In contrast to NbPSD95, which was directly conjugated to the AbberiorStar635P fluorophore, both the r-hcAb and the reference antibody were indirectly detected using identical concentrations of a mouse IgG2-specific secondary nanobody (FluoTag-X2 anti-Mouse IgG2 Ab635P).

In the first set of experiments, all three primary tools were used in primary hippocampal neurons at the molar concentration recommended for the reference antibody (14 nM final concentration). After imaging them in a confocal microscope under the same acquisition conditions, both the nanobody and the r-hcAb showed stronger signals than the reference antibody (Figure 3A and Appendix A). This observation was especially striking as the nanobody-derived signal was comparable to the r-hcAb and even stronger than the control antibody without signal amplification by a secondary reagent. However, when using all primary reagents at only 1 nM final concentration (Figure 3B and Appendix A), even more significant differences became apparent. Here, NbPSD95 showed robust and specific staining, while the signals obtained from the r-hcAb and reference antibody were strikingly dimmer.

In view of the strong signal obtained in confocal microscopy, we decided to challenge NbPSD95 by using a more demanding super-resolution imaging technique, for which probe size and labeling density are crucial. In this respect, fluorescent nanobodies are a preferred alternative to minimize the linkage error between the target molecule and the fluorophore [14,15]. Therefore, we used Stimulated Emission Depletion (STED) microscopy to look at PSD95 on 4% PFA-fixed primary neurons (Figure 4). Remarkably, the strong signal provided by NbPSD95 allowed an effective enhancement in resolution using STED microscopy and even revealed perforated synapses as well as several lateral PSDs (Figure 4), while the confocal image shows uniform structures. The bright and accurately localized signal obtained with NbPSD95 demonstrates its ability to achieve a high labeling density within a single PSD. At the same time, the small dimensions of NbPSD95 combined with the direct and site-specific attachment of fluorophores help minimize the linkage error and signal dispersion associated with conventional polyclonal secondary antibody detection methods [16].

We went one step further and wanted to analyze the performance of NbPSD95 in immunohistochemistry (IHC) applications on PFA-fixed mouse and rat brain sections (Figure 5 and Figure 6). Due to the extremely dense protein assemblies within authentic mature PSDs present in the brain, such staining procedures are generally considered challenging, especially after standard tissue fixation procedures with 4% PFA. When performing the experiment, however, NbPSD95 was immediately able to brightly stain brain features that are typical for PSD95-specific patterns [17,18,19] (Figure 5). As predicted by the strong sequence similarity of mouse and rat PSD95 within PDZ domains 1 and 2 (Figure 1C), corresponding mouse and rat tissue were stained equally well (Figure 5). In line with the sequence identity within the relevant rat and human protein fragments, we have additional evidence that NbPSD95 is also capable of binding human PSD95 present in bioengineered neuronal organoids (BENOs) [20] (Patapia Zafeiriou, University Medical Center Göttingen; personal communication).

When looking at the same tissue preparation by STED super-resolution microscopy (Figure 6), single PSDs could readily be resolved as elongated cone-shaped structures in PFA-fixed authentic brain tissue (Figure 6B). These results show that one-step NbPSD95 fluorescent IHC/ICC staining in fixed brain samples is possible without the need for PSD95-specific fixation or antigen retrieval protocols.

As a final challenge, we tested an anti-PSD95 r-hcAb (NbPSD95 fused to rabbit Fc domains) in IHC on paraffin-embedded brain and retinal sections from mouse and rat (IHC-P; Figure 7). This r-hcAb-based staining successfully revealed defined signals, precisely reproducing the expected localization of PSD95 in the tested tissues. This refers, e.g., to the hippocampi areas rich in synapses between mossy fibers and the pyramidal neurons from CA1/CA3 and the granular neurons within the dentate gyrus. Similarly, a clear staining of the basket cell synapses from the molecular layer communicating with the Purkinje cell somas of the cerebellum was observed [21].

In the retina (Figure 7C), the anti-PSD95 r-hcAb showed the expected prominent staining at the outer plexiform layer (OPL), where large invaginating excitatory synapses from the photoreceptors with bipolar and horizontal cells are present. An additional signal was detected in the inner plexiform layer (IPL). The weaker signal observed in this layer might reflect the fact that, here, inhibitory synapses are mixed with excitatory synapses from the contacts between bipolar and ganglion cells. Our results suggest that NbPSD95 in its r-hcAb format is able to provide a robust and specific staining on formaldehyde-fixed paraffin-embedded sections of various mouse and rat tissues.

## 3. Discussion

Here, we introduce and characterize nanobody-based tools that detect the postsynaptic marker protein PSD95 in mouse and rat tissues. Besides a bona fide nanobody (NbPSD95), which can be site-specifically conjugated to fluorophores (termed FluoTag-X2 anti-PSD95) or to labels like, e.g., biotin or DBCO, we also created a range of nanobody fusions to various IgG Fc domains (termed r-hcAbs). As all tools are based on the identical specificity-determining moiety, it is expected that the reagents will recognize the same epitope. While immunization was done using a PSD95 fragment comprising the PDZ-domains 1 and 2 from mice, our results show that the NbPSD95 works also on rat PSD95. Additionally, we have confirmation that this nanobody binds specifically PSD95 from human-derived organoids [20] (personal communication; Patapia Zafeiriou, University Medical Center Göttingen), which is in line with the high degree of sequence conservation between rat, mouse, and human PSD95 within the protein fragment used for immunization.

Importantly, we could show in various assays that NbPSD95 specifically recognizes PSD95; we could not find any significant interaction with relevant fragments from related PDZ-domain containing proteins like SAP97, PSD93, and SAP102. In accordance with these results, in Western blot analyses, the PSD95-specific signal was entirely missing in a PSD95 knock-out lysate.

NbPSD95 and NbPSD95-derived r-hcAbs proved ideal for different experimental setups. Directly labeled NbPSD95 was a superior tool for all applications performed on PFA-fixed samples. Importantly, and in contrast to most established reagents for detecting PSD95, NbPSD95 was fully proficient in brightly staining tissues subjected to standard fixation conditions using 4% PFA without further antigen retrieval steps. At the same time, due to its small size and monovalent binding mode, this directly labeled nanobody format is ideal for quantitative imaging and super-resolution microscopy, as additional non-linear amplification effects and signal blurring due to bulky secondary reagents are effectively circumvented [16]. In contrast, the anti-PSD95 r-hcAbs proved to be superior tools for applications on denatured protein samples or samples subjected to antigen retrieval after paraffin embedding.

Interestingly, NbPSD95 (~15 kDa) and the corresponding r-hcAb (~80 kDa) showed clear differences in their staining intensities (see Figure 3). As both probes detect the same epitope, these differences can most likely be attributed to the larger size of the r-hcAb, which restricts access to the protein-dense environment of the PSD. Such a size effect might also explain why, at the same concentration, the yet larger control antibody (~150 kDa) basically failed to produce consistent and bright staining (probably only detecting a subset of PSD95 targets). This interpretation is in line with the observation that the nanobody, as the smallest binder, was able to robustly stain the target structure both at high and low concentrations in samples fixed with 4% PFA even without the need to perform further PSD95-specific processing or antigen retrieval steps. Our direct comparison also corroborates the common observation that the detection of PSD95 remains challenging when using conventional antibodies [11]. As only a subset of PSD95 can be reached by such conventional antibodies, immunostainings performed with such tools often systematically fail to reproduce the authentic localization of PSD95. In comparison, images obtained with both NbPSD95 and NbPSD95-derived r-hcAbs are remarkably well in line with the non-biased expression pattern of PSD95 observed in knock-in mouse lines expressing PSD95 fused to mVenus or HaloTag at endogenous levels [17,19].

Even at the fine structural level, NbPSD95 and its r-hcAb derivative provided an accurate representation of the quantity and location of PSD95 in the crowded synaptic region without the need to use genetically modified animals. This is also suggested by a recent preprint by Shahib et al. using expansion microscopy [22]. Especially when combining super-resolution microscopy with immunohistochemistry, the intricate details observed with NbPSD95 are superior to other available affinity probes targeting PSD95.

The here-described novel set of nanobody-based affinity probes targeting PSD95 is unique as it allows a quantitative and spatially accurate high-resolution localization of PSD95 in various tissues without any special treatment. These tools might therefore be important for synaptic research in general but may also help to resolve the divergent discussions on the presynaptic localization of PSD95 at the ultrastructural level within the retina and cerebellum [23,24].

## 4. Materials and Methods

### 4.1. Recombinant Proteins

Amino acids 66-251 of mouse PSD95 were produced in *E. coli* and fused to an N-terminal His_6_ tag. For that, BL21 transformed with the respective expression vector was cultured at 37 °C under antibiotic selection in 2YT medium. At an OD_600_ of 0.3, expression was induced by the addition of 0.5 mM IPTG, and cells were further cultured for 4 h. Cells were harvested by centrifugation and lysed by sonication in low salt buffer (LS; 50 mM Tris-HCl, 0.3 M NaCl, 5 mM EDTA, pH 7.5) supplemented with 2 mM imidazole. After the removal of cell debris, fusion proteins were captured on an IMAC resin. The resin was washed with >5 column volumes (CV) with an LS buffer containing 300 mM imidazole. The target protein was further purified by gel filtration on a Superdex 75 column equilibrated with PBS.

PDZ domains 1/2 of rat PSD95, PSD93, SAP97, and SAP102 fused to a C-terminal maltose-binding protein (MBP) were expressed in *E. coli* as fusions to an N-terminal His-SUMO tag [25]. In brief, an *E. coli* expression strain transformed with the respective expression vector was cultured at 30 °C under antibiotic selection in a terrific broth (TB) medium. At an OD_600_ of ~4, expression was induced by the addition of 0.4 mM IPTG, and cells were further grown overnight at 23 °C. Cells were harvested by centrifugation and lysed by sonication in buffer high salt (HS; 50 mM Tris-HCl, 1.5 M NaCl, 5 mM EDTA, 15 mM imidazole, pH 7.5). After the removal of cell debris, fusion proteins were captured on an IMAC resin. The resin was washed with >5 column volumes (CV) of HS buffer followed by >5 column volumes of LS buffer containing 15 mM imidazole. Target proteins were eluted by incubation with 100 nM bdSENP1 protease [25] for 1 h at 4 °C. The eluted proteins were further purified by gel filtration on a Superdex 200 column equilibrated with PBS.

### 4.2. Immunization and sdAb Selection

Two male alpacas between 2 and 3 years old were immunized six times, each with 500 µg of recombinant protein comprising amino acids 66-251 of mouse PSD95 mixed with GERBU adjuvant F (#3030) as suggested by the manufacturer. Five days after the last immunization, PBMCs were extracted and total RNA purified. SdAbs-encoding cDNA was amplified by a 3-step nested RT-PCR protocol and cloned into a pHEN2-derived phagemid [12]. Recombinant phages displaying target-specific sdAbs were enriched in three consecutive panning rounds by phage display. 96 individual clones from the enriched pool of target-specific sdAbs were purified and screened by a single-clone ELISA to assess the monovalent affinity of each clone and for specific binding to the immunized target.

### 4.3. ELISA Assays

Recombinantly expressed proteins comprising PDZ domains 1/2 of rat PSD95, PSD93, SAP97, or SAP102 fused to a C-terminal MBP were coated overnight at 4 °C on ELISA plates (Nunc Maxisorb Immunoplates, #442404, Thermo, Waltham, MA, USA) at a concentration of 100 ng per well. After blocking with SmartBlock (#113500, Candor Biosciences, Wangen, Germany) for 1 h at room temperature (RT) and washing (3 × 5 min in TBS-T), the plates were incubated with 100 nM of each purified clone (single-clone ELISA) or a dilution series of NbPSD95 fused to a C-terminal 3xFLAG-GFP tag for 1 h at RT. Plates were washed as before and incubated with an HRP-conjugated anti-FLAG antibody (clone M2, Sigma-Aldrich, Burlington, MA, USA, 1:10.000 dilution) for 1 h at RT. After a final washing step, plates were developed with SeramunBlau slow2 50 (#S150-TMB, Seramun, Heidesee, Germany) for 5 min at RT before stopping the reactions with the addition of H_2_SO_4_ (0.5 M final). Plates were scanned using a SpectraMax M2 plate reader (Molecular Devices, San Jose, CA, USA).

### 4.4. Transfection Constructs

For eukaryotic expression of EGFP-tagged PDZ domains 1/2 of rat PSD95, PSD93, SAP97, and SAP102 on mitochondria, the respective PDZ domains were cloned in-frame into a vector encoding an N-terminal TOM70-derived signal sequence and EGFP under the control of a CMV promoter.

### 4.5. Antibodies

Antibodies used in the current study are summarized in Table 1. Concentrations/dilutions and further experimental details for each experiment are given in Appendix A.

### 4.6. Western Blots

PSD95 wild type and knock-out lysates for Western blotting were courtesy of Prof. Seth Grant, University of Edinburgh. Total brains from PSD95 wild type and knock-out tissues were homogenized and lysed in SDS sample buffer [26]. Lysates were resolved by SDS-PAGE (15 µg of total protein per lane) and transferred to a nitrocellulose membrane. After blocking with 5% milk powder in TBS-T, membranes were incubated with a mouse anti-beta-actin antibody (Synaptic Systems; #251 011, Göttingen, Germany) followed by an HRP-coupled secondary nanobody (sdAb anti-Mouse IgG1 HRP, NanoTag Biotechnologies; #N2005-HRP) for detection of beta-actin used as a loading control. PSD95 was detected by three different anti-PSD95 r-hcAbs (recombinant anti-PSD95 antibodies, NanoTag Biotechnologies; Mouse IgG1 Fc fusion #N3782, Mouse IgG2 Fc fusion #3785, and Rabbit IgG Fc fusion #3783). After washing (3 × 5 min in TBS-T), the r-hcAbs were detected using a corresponding HRP-coupled secondary nanobody (NanoTag Biotechnologies, sdAb anti-Mouse IgG1 HRP #N2005-HRP, sdAb anti-Mouse IgG2 HRP #N2705-HRP, or sdAb anti-Rabbit IgG HRP # N2405-HRP, respectively). Blots were developed using Pierce ECL Western Blotting Substrate (Pierce, #32209, Boston, MA, USA). For antibody details, see Appendix A.

### 4.7. Cell Culture

COS-7 cells were cultured in Dulbecco’s MEM supplemented with 10% fetal bovine serum, 2 mM L-glutamine, 1% penicillin, and streptomycin. Cells were cultured at 37 °C 5% CO_2_ in a humidified incubator. For immunostainings, cells were plated on poly-L-lysine (PLL)-coated coverslips. Rat primary hippocampal neuron cultures for imaging were prepared from the brains of P1-2 rat pups. The hippocampi were extracted and placed in a solution containing 10 mL of DMEM (Thermo Fisher, Waltham, MA, USA), 1.6 mM cysteine, 1 mM CaCl_2_, 0.5 mM EDTA, and 25 units of papain per mL of solution, with CO_2_ bubbling, at 37 °C for 1 h. The hippocampi were triturated using a serological pipette in complete neurobasal medium (Neurobasal A (Thermo Fisher, Waltham, MA, USA), containing 2% B27 (Thermo Fisher, Waltham, MA, USA) and 1% Glutamax-I (Thermo Fisher, Waltham, MA, USA). Neurons were plated on glass coverslips coated with poly-L-lysin hydrochloride (1 mg/mL, Sigma-Aldrich, Burlington, MA, USA). After 2 h in the incubator, the plating medium was replaced with 1.25 mL of complete neurobasal medium, and cells were incubated for 15 days at 37 °C with 5% CO_2_ in a humidified incubator.

### 4.8. Immunofluorescence

COS-7 cells and primary hippocampal neurons were fixed using 4% PFA in PBS at pH ~7.5 for 30 min at RT. After removing PFA, the remaining aldehydes were quenched using 0.1 M of NH_4_Cl or 0.1 M glycine prepared in PBS for 15–20 min. Blocking and permeabilization of cells were performed by incubating the slides twice for 10–15 min each with gentle orbital shaking in PBS supplemented with 2% BSA and 0.1% Triton X-100.

For Figure 2B,D and Figure 4, transfected COS-7 cells or neurons were incubated with the indicated directly labeled nanobodies (see Appendix A) for 1 h with orbital shaking, followed by 3 washing steps of 5 min each using PBS. Cells were stained with DAPI before mounting with Mowiol (12 mL of 0.2 M Tris buffer pH 8.5, 6 mL distilled water, 6 g glycerol, 2.4 g Mowiol 4-88, Merck Millipore, Burlington, MA, USA).

For Figure 3 and Appendix A, all fixed primary neuronal cultures were simultaneously stained with fluorescently labeled nanobodies recognizing GFAP and Synaptotagmin 1 for 1 h at RT with orbital shaking. The detection of PSD95 was performed in three alternative setups: In the first line of experiments, PSD95 was detected using a directly labeled NbPSD95 (1 nM or 14 nM final concentration, as indicated). For indirect detection of PSD95, neurons were first incubated with equivalent concentrations (1 nM and 14 nM final concentrations each) of an anti-PSD95 r-hcAb (NbPSD95 fused to a mouse IgG2 Fc domain) or an established mouse IgG2 monoclonal antibody recognizing PSD95. In these indirect setups, after washing (3 × 5 min in PBS each), the anti-PSD95 r-hcAb or antibody was detected using a fluorescently labeled secondary nanobody recognizing the mouse IgG2 Fc domain present in both the r-hcAb and the antibody. After washing (3 × 5 min, PBS) and DAPI staining of nuclei, samples were mounted in Mowiol for imaging. See Appendix A for more details on the antibodies used.

### 4.9. Immunohistochemistry

Mice and rats were transcardially perfused with 0.9% saline, followed by 4% formaldehyde. Brains were removed, post-fixed for 24 h with 4% formaldehyde, and cut into 50 µm thick sections on a vibratome. Sections were stored in a cryoprotection solution (25% ethylene glycol and 25% glycerol in PBS) at −20 °C until staining.

For the staining shown in Figure 5 and Figure 6, sections were permeabilized and blocked for 1 h at RT with 10% normal goat serum (NGS), 0.3% Triton X-100 in TBS. Brain sections were incubated overnight at 4 °C with indicated antibodies and/or nanobodies (see Appendix A) diluted in 5% NGS, 0.3% Triton X-100 in TBS. After washing, sections were counterstained with DAPI and mounted with Entellan.

### 4.10. IHC-P

Brains and eyes were prepared from transcardially perfused mice and rats and post-fixed with 10% (*v*/*v*) neutral buffered formalin for 24 h at 4 °C. Dehydration, clearing, and paraffin infiltration were performed in an automated tissue processing system. Paraffin blocks were prepared and cut into 3.5 µm thick tissue sections. Tissue sections were deparaffinized and subjected to antigen retrieval in Tris-EDTA buffer (10 mM Tris, 1 mM EDTA, 0.05% Tween 20, pH 9.0) for 30 min at 97–100 °C. After inhibition of endogenous peroxidase activity, sections were blocked in a serum-free protein block (X0909, Agilent, Santa Clara, CA, USA) for 10 min at RT. Then, sections were incubated with 1 µg/mL recombinant anti-PSD95 antibody (Rabbit Fc fusion, NanoTag Biotechnologies #N3783, Göttingen, Germany) diluted in Dako REAL Antibody Diluent (S2022, Agilent) for 1 h at RT. A biotinylated anti-Rabbit antibody was added to the slides for 30 min at RT. An avidin-biotin complex (ABC)-HRP Kit (PK-4000, Vector Laboratories, Burlingame, CA, USA) was applied for a further 30 min at RT, and ImmPACT DAB substrate (SK-4105, Vector Laboratories, Burlingame, CA, USA) was used to visualize antibody binding. Nuclei were counterstained with hematoxylin solution. Slides were mounted in an organic mounting medium (Entellan, 1.079.610.500, Sigma-Aldrich, Burlington, MA, USA). Details on the antibodies used are summarized in Appendix A.

### 4.11. Microscopy

Chromogenic stained images were obtained with an Axiolab 5 brightfield microscope (Carl Zeiss GmbH, Jena, Germany) equipped with an Axiocam 208 color microscope camera. Images were processed with the ZEN software 3.5. Epifluorescence microscopy images were obtained with an Axio Observer 7 (Carl Zeiss GmbH) equipped with an Axiocam 807 and an Apotome 3. Pictures were taken with 2.5 × 0.12 NA dry and 63 × 1.4 NA oil immersion lenses and processed with the ZEN software 3.6. For confocal and STED microscopy, images were acquired using the STED Expert line microscope (Abberior Instruments, Göttingen, Germany). The scope body was an IX83 inverted microscope (Olympus, Hamburg, Germany) equipped with an UPLSAPO 100 × 1.4 NA oil immersion objective (Olympus). The 488 nm, 561 nm and 640 nm lasers were used for confocal imaging. Stimulated Emission Depletion images were obtained using the 775 nm pulsed STED depletion lasers in combination. Images were analyzed and colorized using Fiji/ImageJ (v. 1.53o).

## Figures and Tables

**Figure 1 ijms-24-07294-f001:**
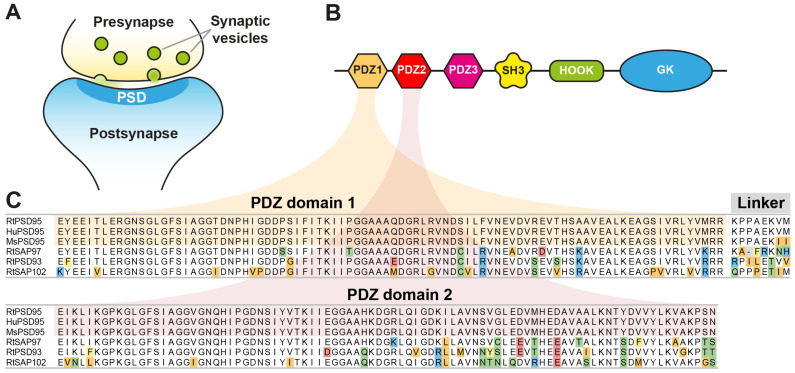
Localization, domain structure and sequence of PSD95. (**A**) A sketch of an excitatory synapse. PSD95 is a major scaffold constituent of the post-synaptic density (PSD), localized juxtaposed to the synaptic vesicle release site. (**B**) Putative domain structure of PSD95: PDZ1/2/3, SH3, HOOK, and the non-catalytic Guanylate Kinase domain (GK). (**C**) Sequences of PDZ domains 1 and 2 of rat (Rt), human (Hu), and mouse (Ms) PSD95, as well as the most closely related proteins PSD93, SAP97, and SAP102 from rat. Within this sequence stretch, the rat and human PSD95 sequences are identical and differ by only two amino acids in the linker connecting PDZ1 and PDZ2 from the respective mouse sequence.

**Figure 2 ijms-24-07294-f002:**
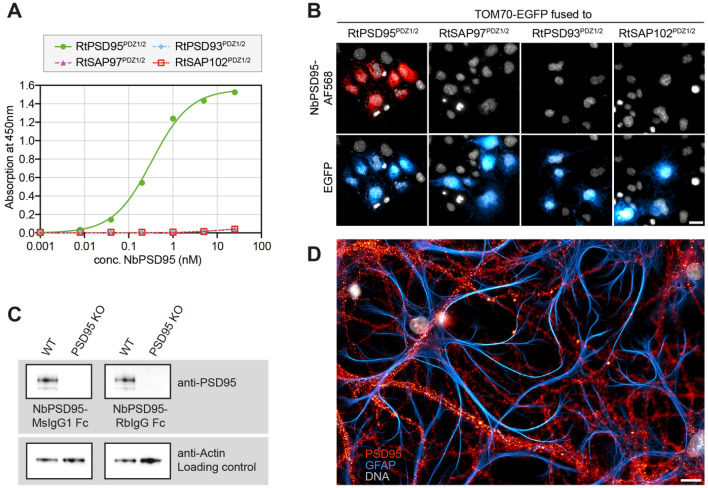
NbPSD95 specifically recognizes PSD95. (**A**) A dilution series of FLAG-tagged NbPSD95 was applied to ELISA plates coated with indicated PDZ domains fused to *E. coli* maltose-binding protein (MBP). The detection was performed with an HRP-coupled anti-FLAG antibody. A significant signal could only be obtained for PSD95^PDZ1/2^. (**B**) COS-7 cells were transfected with the indicated PDZ-domains fused to EGFP and a mitochondrial targeting sequence. After fixation with 4% PFA, the cells were stained with NbPSD95 coupled to AZDye568 (structurally identical to AlexaFluor568). The co-localization of NbPSD95 with GFP required the presence of PSD95^PDZ1/2^. Scale bar: 20 µm. (**C**) Total brain lysates from a wild-type (WT) mouse or a PSD95 knock-out mouse (PSD95 KO) were analyzed by Western blot using r-hcAb comprising NbPSD95 and a Fc domain from either mouse IgG1 (MsIgG1 Fc; left) or rabbit IgG (RbIgG Fc; right). Actin served as a loading control. Both r-hcAbs readily detected PSD95 in the WT lysate, while no signal was obtained using the PSD95 KO lysate, indicating that the r-hcAbs exclusively recognize PSD95. (**D**) PFA-fixed hippocampal primary neuronal cultures from rats were stained with NbPSD95 coupled to AZDye568 (false color representation in red). Counterstaining was performed with an anti-GFAP nanobody (FluoTag-X2 anti-GFAP Atto488, false color representation in cyan). NbPSD95 produced the typical punctate staining on neuronal structures. Scale bar: 20 µm.

**Figure 3 ijms-24-07294-f003:**
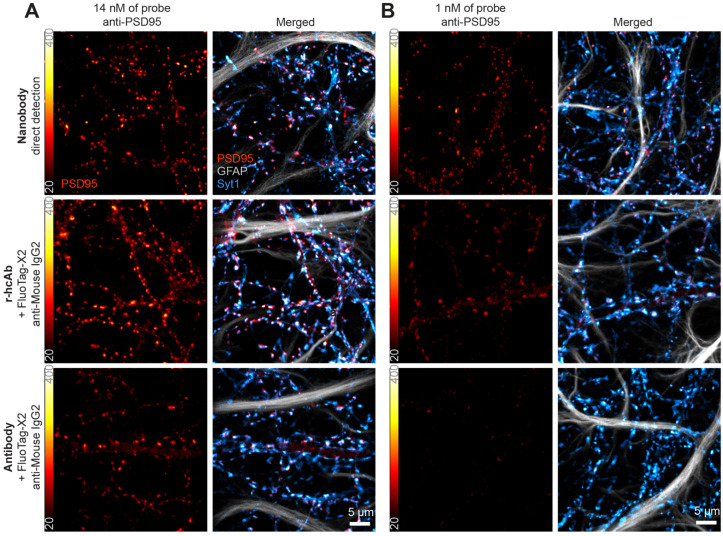
Comparison of NbPSD95 with an anti-PSD95 r-hcAb and a conventional antibody. Primary hippocampal neurons were stained with NbPSD95 directly labeled with AbberiorStar635P (nanobody), with an anti-PSD95 r-hcAb harboring a mouse IgG2 Fc domain (r-hcAb) or an established mouse monoclonal IgG2 antibody (antibody), all at a concentration of 14 nM (**A**) or 1 nM (**B**). Identical concentrations of FluoTag-X2 anti-Mouse IgG2 coupled to AbberiorStar635P were used as secondary reagents for the detection of both the r-hcAb and the mouse monoclonal antibody. Merged figures show the respective anti-PSD95 signal (red) together with counter-stainings performed with FluoTag-X2 anti-GFAP conjugated to Atto488 (grey) and FluoTag-X2 anti-Syt1 (synaptotagmin 1) conjugated to AZDye568 (cyan). Confocal images were acquired with equal settings for each channel and images equally scaled to allow a direct comparison. Separate channels are shown in Appendix A.

**Figure 4 ijms-24-07294-f004:**
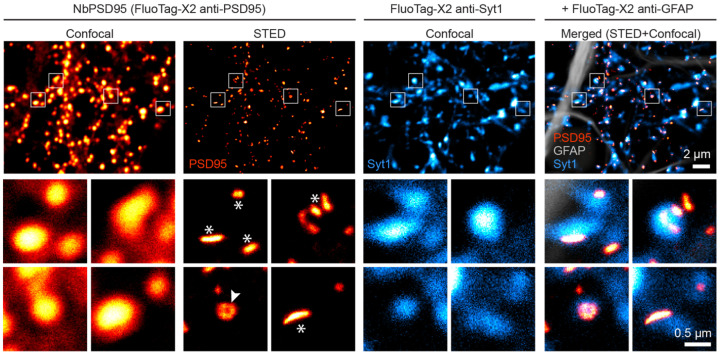
Confocal and STED imaging on primary rat hippocampal neurons. Primary hippocampal culture neurons were stained using NbPSD95 (FluoTag-X2 anti-PSD95, red) conjugated to AbberiorStar635P and imaged using Confocal and STED microscopy. For counterstaining, the pre-synaptic protein Synaptotagmin 1 (Syt1; cyan) and glia marker GFAP (grey) were imaged in confocal mode for reference. Four selected regions denoted by white squares are magnified, displaying conventional lateral PSD shapes (asterisks) as well as so-called perforated synapses (arrowheads) only visible under STED microscopy.

**Figure 5 ijms-24-07294-f005:**
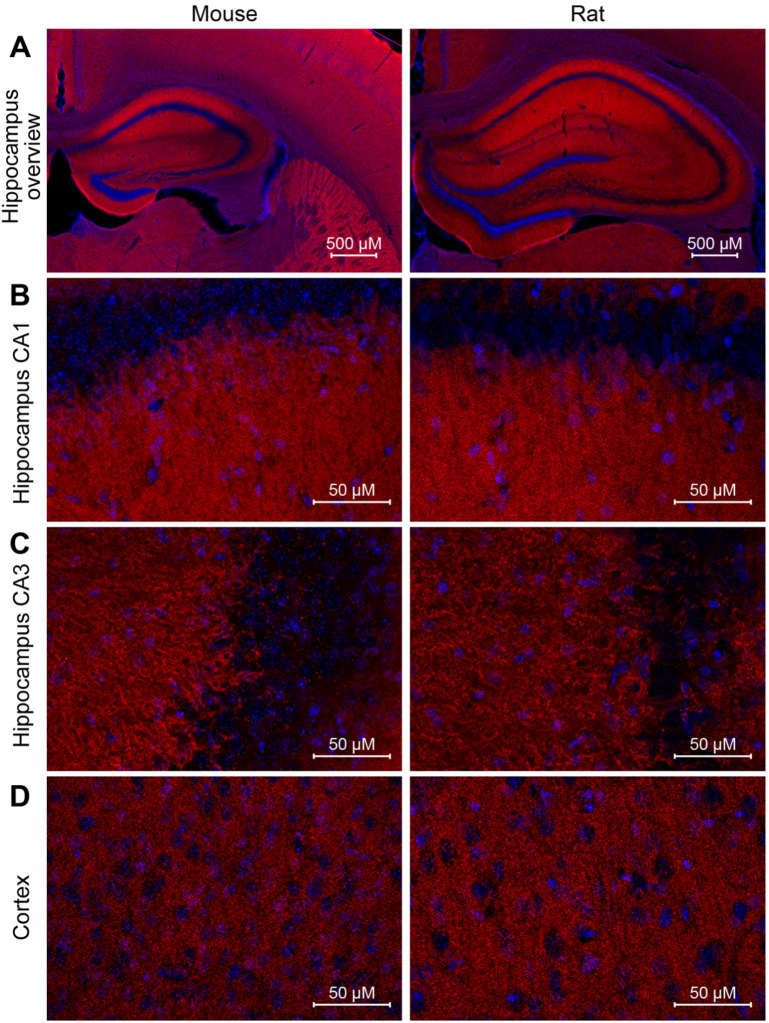
Immunohistochemistry using NbPSD95 on formalin-fixed brain slices. Mouse (left) and rat (right) brains were fixed in 4% formaldehyde for 24 h and cut into 50 µm thick sections using a vibratome. Sections were stained with Sulfo-Cy3-conjugated NbPSD95 (red) and DAPI (blue). Without antigen retrieval, the nanobody reveals PSD95-positive synapses in the mouse and rat brains. (**A**) Epifluorescence imaging of the mouse and rat hippocampus shows the expected intensity pattern with strong and dense staining in the stratum oriens and stratum radiatum of CA1. (**B**–**D**) Higher magnification images show numerous synaptic dots in the hippocampal regions of CA1 (**B**) and CA3 (**C**) or in the cortex (**D**). The staining pattern suggests a high specificity for NbPSD95 towards excitatory synapses. In all regions, NbPSD95 works comparably well in mouse and rat specimens.

**Figure 6 ijms-24-07294-f006:**
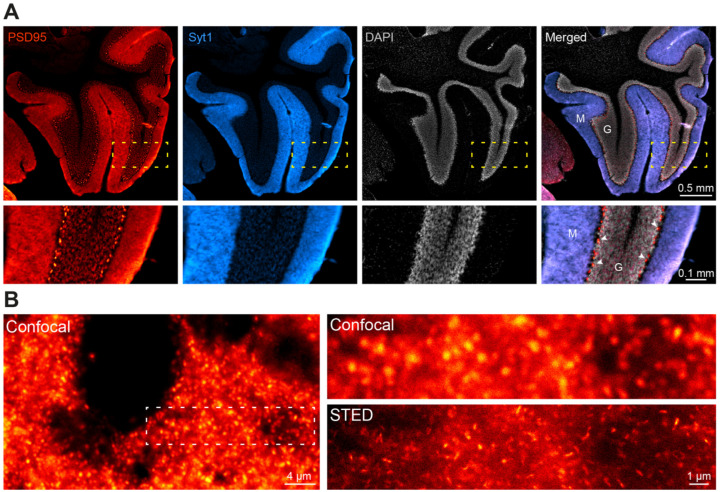
Super-resolution microscopy on formalin-fixed mouse cerebellar cortex. Brains were fixed in 4% formaldehyde for 24 h and cut into 50 µm thick sections using a vibratome. (**A**) Epifluorescence imaging overview of a cerebellar cortex section (top) and a zoom-in (bottom) on the granular cell layer (G) and molecular layer (M). AbberiorStar635p-conjugated NbPSD95 (red) was used for revealing post-synapses, and an antibody directed against Synaptotagmin 1 was used as a marker of pre-synapses (cyan). DAPI staining (gray) is only visible within the granular cell layer, while a majority of the staining by NbPSD95 and the anti-Syt1 antibody are found on the molecular layer. In addition, NbPSD95 gives a bright signal in the Purkinje cell layer at the axosomatic synapses directly contacting the cell body of Purkinje cells (arrowheads). (**B**) high-magnification imaging using laser scanning confocal and STED imaging on the molecular layer (M). The signal obtained with NbPSD95 is sufficiently strong to even support enhanced resolution using STED microscopy on tissue.

**Figure 7 ijms-24-07294-f007:**
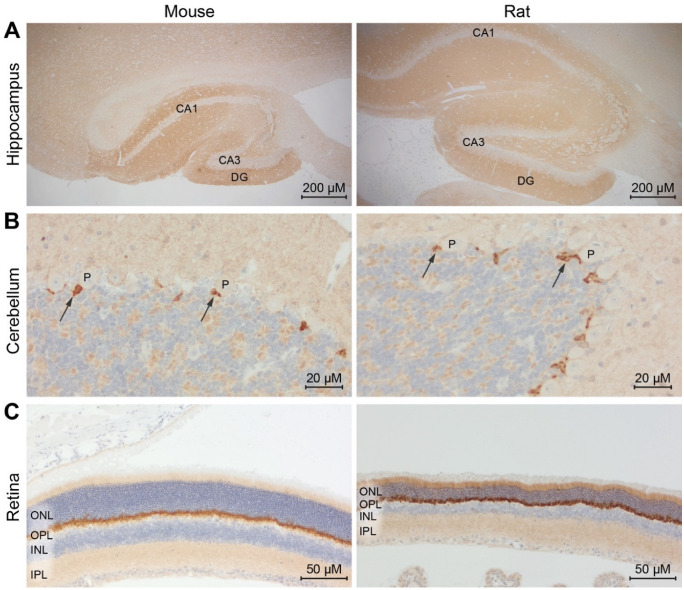
IHC-P using an anti-PSD95 r-hcAb. Chromogenic IHC staining of an anti-PSD95 r-hcAb in formalin-fixed paraffin-embedded (FFPE) mouse (left) and rat (right) tissue sections using 3,3′-Diaminobenzidin (DAB; brown staining). Hematoxylin provides blue nuclear staining. (**A**) Immunohistochemistry using the rabbit anti-PSD95 r-hcAb reveals the characteristic distribution of PSD95 in coronal sections of mouse and rat hippocampi. All regions of the hippocampus are labeled. However, DAB staining is more intense in the pyramidal cell layer of the CA1 area and in the molecular layer of the dentate gyrus (DG). (**B**) High-magnification views of mouse and rat FFPE cerebellum sections. Strong PSD95 staining is detected within the basket cell terminal pinceaux (arrows). Purkinje cells (P) and neurons in the granule layer are not stained. (**C**) In FFPE sections of mouse and rat retina, very intense DAB staining is present in the outer plexiform layer (OPL). Moderate punctate staining is observed in the inner plexiform layer (IPL). The inner and outer nuclear layers (INL and ONL) are not stained.

**Table 1 ijms-24-07294-t001:** Antibodies.

Product	Product Number	Provider
FluoTag-X2 anti-PSD95 AZDye568	N3702-AF568-L	NanoTag Biotechnologies
FluoTag-X2 anti-PSD95 Sulfo-Cy3	N3702-SC3-L	NanoTag Biotechnologies
FluoTag-X2 anti-PSD95 AbberiorStar635P	N3702-Ab635P-L	NanoTag Biotechnologies
FluoTag-X2 anti-Syt1 AZDye568	N2302-AF568-L	NanoTag Biotechnologies
Recombinant anti-Synaptotagmin antibody	105 008	Synaptic Systems
FluoTag-X2 anti-Syt1 Sulfo-Cy3	N2302-SC3-L	NanoTag Biotechnologies
FluoTag-X2 anti-GFAP Atto488	N3802-At488-L	NanoTag Biotechnologies
FluoTag-X2 anti-Mouse IgG2 AbberiorStar635P	N2702-Ab635P	NanoTag Biotechnologies
FluoTag-X2 anti-Rabbit IgG AbberiorStar580	N2402-Ab580-L	NanoTag Biotechnologies
Recombinant anti-PSD95 Antibody, Rabbit Fc fusion	N3783	NanoTag Biotechnologies
Recombinant anti-PSD95 Antibody, Mouse IgG1 Fc fusion	N3782	NanoTag Biotechnologies
Recombinant anti-PSD95 Antibody, Mouse IgG2 Fc fusion	N3785	NanoTag Biotechnologies
Secondary anti-Rabbit Biotin	111-065-144	Jackson Immuno Research
Avidin-Biotin Complex (ABC)-HRP Kit	PK-4000	Vector Laboratories
Mouse anti-beta-Actin	251 011	Synaptic Systems
Monoclonal anti-PSD95 antibody	MABN68	Merck Millipore
sdAb anti-RbIgG HRP	N2405-HRP	NanoTag Biotechnologies
sdAb anti-MsIgG1 HRP	N2005-HRP	NanoTag Biotechnologies
sdAb anti-MsIgG2 HRP	N2705-HRP	NanoTag Biotechnologies
Monoclonal anti-FLAG-HRP	A8592	Merck Millipore

## Data Availability

Data of Figures presented in this study is available upon reasonable requests.

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
