# Peer review of "Simple and Highly Efficient Detection of PSD95 Using a Nanobody and Its Recombinant Heavy-Chain Antibody Derivatives"

_ijms, 2023, doi:10.3390/ijms24087294_

Round 1

Reviewer 1 Report

Kilisch et al developed and characterizes camelid based PSD95 single-domain antibody (nanobody) that allows for a more precise detection and better quantification of PSD95 to rat, mouse, and human samples in the crowded synaptic regions. Importantly, directly labeled NbPSD95 worked well with PFA-fixed samples and its small size extend its application to quantitative imaging and super-resolution microscopy. Overall, these minibodies are convincingly superior tools for applications on denatured protein samples or samples subjected to antigen retrieval after paraffin embedding which is a big deal to the field on both the fronts (i) as a nice tool and (ii) therapeutic intervention. The data are well represented, and image quality is impressive, and it is well written. I have minor comments-

1.     In method section include the age of alpacas and dose of recombinant protein used for immunization. Did authors used any adjuvant as well? After how many days of immunization PBMCs were isolated?

2.     Line 280-281: Improve sentence.

3.     Line 288: I am confused with the sentence PSD95-specific signal was entirely in a PSD95 knock-out lysate. What does it mean?

Author Response

Point 1: In method section include the age of alpacas and dose of recombinant protein used for immunization. Did authors used any adjuvant as well? After how many days of immunization PBMCs were isolated?

Response 1: We have added in the Method section more details regarding the alpacas, immunization procedure, including adjuvant and timepoint of blood draw for PBMC isolation.

Point 2: Line 280-281: Improve sentence.

Response 2: We have reworded the sentence.

Point 3. Line 288: I am confused with the sentence PSD95-specific signal was entirely in a PSD95 knock-out lysate. What does it mean?

Response 3: We have corrected the sentence.

Reviewer 2 Report

In this paper not the authors identified a Nb directed against mouse PDZ1-PDZ2 of PSD95. This Nb is used in high resolution imaging of the target in various samples. 

The data are clearly presented, well described and convincing. 

A few minor points of attention:

- The term minibody to denote the Nb fused to the hinge-CH2-CH3 is confusing as minibody has been attributed to Nb (or scFv) fused to CH3 this construct is spontaneously dimerising to form a bivalent construct; Why don't you use hcAb instead of minibody for this chimeric construct?

-Line 94 "a pHen2 derived phagemid" while in Materials and Methods it is written 'pHen3-derived' (Line 358). Nowhere the reader can find a reference to what pHEN2 or pHen3 are or in what they differ from each other. Could you refer to a previously published paper? 

- Line 97: "The anti-PSD95 nanobody clone 1B2 displayed the strongest reactivity". This is fine however, the reader has no clue of what has been done: what was the amount of antigen used during the 3 rounds of panning (always the same amount, decreasing amounts, stringency of washing...?) Apparently 96 clones were tested. How many were positive, and what do you mean by "strongest reactivity". Probably you ar referring to the signal intensity, but this is only reliable if you use purified Nbs with known starting concentration. If you start with crude lysate or phage displayed material then the signal intensity is a function of the affinity (or possibly avidity when working with virions) and the amount of functional (properly folded and soluble) Nb in the extract. Thus, this is not a good manner to identify the "strongest reactivity". The description in M&M lines 359-361 is vague ("enriched poo of target-specific sdAbs was screened by ELISA for specific binding to the immunised target.").

- The reactivity of the Nb in WB is surprising as Nbs are most of the time lousy reagents in WB since they normally recognise conformational epitopes that are disrupted during SDS gel electrophoresis. Therefore, I was wondering how much material was applied on the SDS-PA gel? 

-Line 301: the reconstituted hcAb has a MW around 80 kDa "the corresponding minibody (-809 kDa)". However, in supplementary material the SDS-PAGE reveals a MW above 100 kDa. This is something that occurs regularly: the reconstituted monoclonal hcAb has a theoretical MW of around 90 kDa, but on SDS-PAGE it runs with a MW of around 110 kDa. 

-Line 302-303: "As both probes comprise identical antigen-binding moieties". This statement is not correct. Both probes comprise the same antigen binding molecules, however, for the Nb it is in a monovalent form while in the hcAb it is in a bivalent format, which can have widely different behaviours due to size differences and affinity versus avidity effects. 

- Confusion in material and methods about buffer LS: is that the same as lysis buffer ?(line 339)

Author Response

Point 1: The term minibody to denote the Nb fused to the hinge- CH2-CH3 is confusing as minibody has been attributed to Nb (or scFv) fused to CH3 this construct is spontaneously dimerising to form a bivalent construct; Why don't you use hcAb instead of minibody for this chimeric construct?

Response 1: We have changed the terminology throughout the text. We have now termed it a recombinant heavy-chain antibody (r-hcAb) to differentiate from the naturally occurring heavy-chain antibodies (IgG2 & IgG3) in camelids.

Point 2: Line 94 "a pHen2 derived phagemid" while in Materials and Methods it is written 'pHen3-derived' (Line 358). Nowhere the reader can find a reference to what pHEN2 or pHen3 are or in what they differ from each other. Could you refer to a previously published paper?

Response 2: We have corrected the typo to pHEN2 and added the corresponding reference on Material and Methods.

Point 3: Line 97: "The anti-PSD95 nanobody clone 1B2 displayed the strongest reactivity". This is fine however, the reader has no clue of what has been done: what was the amount of antigen used during the 3 rounds of panning (always the same amount, decreasing amounts, stringency of washing...?) Apparently 96 clones were tested. How many were positive, and what do you mean by "strongest reactivity". Probably you ar referring to the signal intensity, but this is only reliable if you use purified Nbs with known starting concentration. If you start with crude lysate or phage displayed material then the signal intensity is a function of the affinity (or possibly avidity when working with virions) and the amount of functional (properly folded and soluble) Nb in the extract. Thus, this is not a good manner to identify the "strongest reactivity". The description in M&M lines 359-361 is vague ("enriched poo of target-specific sdAbs was screened by ELISA for specific binding to the immunised target.").

Response 3: The sentence in Line 97 states that NbPSD95 provided the strongest signal from a single-clone ELISA assay (now better explained in Methods). 100 ng of antigen was immobilized per well and purified sdAb clones were used with a known concentration (100 nM) for the ELISA. Thus, the results among selected clones are directly comparable. We do not see the need to describe how many “positive” and “negative” clones were found, since this is not a binary result, but rather a gradient of colorimetric signals, and ultimately, we follow the characterization of the best-performing clone candidate (i.e., the one providing the strongest colorimetric signal when reading the ELISA results).

We have expanded the explanation on lines 359-360, but please notice that the detailed (also extended) ELISA procedure is described in section 4.3 on M&M.

Point 4: The reactivity of the Nb in WB is surprising as Nbs are most of the time lousy reagents in WB since they normally recognise conformational epitopes that are disrupted during SDS gel electrophoresis. Therefore, I was wondering how much material was applied on the SDS-PA gel?

Response 4: We have added in the M&M the amount loaded per lane in the WB (15 µg total brain lysate proteins).

Point 5: Line 301: the reconstituted hcAb has a MW around 80kDa "the corresponding minibody (-80.9 kDa)". However, in supplementary material the SDS-PAGE reveals a MW above 100 kDa. This is something that occurs regularly: the reconstituted monoclonal hcAb has a theoretical MW of around 90 kDa, but on SDS-PAGE it runs with a MW of around 110 kDa.

Response 5: The WB on Supp. Fig 1 corresponds to the detection of PSD95 from rat brain lysates using the different recombinant hcAb (i.e., NbPSD95 fused to Ms IgG1 Fc, Ms IgG2 Fc or Rb IgG Fc domain), and not to the MW of the r-hcAbs. The aim of this Figure is to demonstrate that regardless of the chimeric Fc domain added to the NbPSD95 the antigen specificity is retained.

Point 6: Line 302-303: "As both probes comprise identical antigen-binding moieties". This statement is not correct. Both probes comprise the same antigen binding molecules, however, for the Nb it is in a monovalent form while in the hcAb it is in a bivalent format, which can have widely different behaviours due to size differences and affinity versus avidity effects.

Response 6: We have rephrased it to “As both probes detect the same epitope, these differences can most likely be attributed to the larger size of the r-hcAb, which restricts the access to the protein-dense environment of the PSD”.

Monovalent and divalent binding effects are discussed in the previous paragraph on the main text; in this paragraph (on lines 302-303), we discuss that in Fig 3, the physical size of the probe, seems to play a role in finding epitopes in a crowded environment like the PSD.

Point 7: Confusion in material and methods about buffer LS: is that the same as lysis buffer? (line 339)

Response 7: We have clarified this issue.
